# OpenReview forum: "Multi-Scale Adaptive Hypergraph Learning for High-Order Brain Connectivity Analysis"
_ICLR.cc/2026/Conference — Submitted to ICLR 2026_

### Official Review · Reviewer_dwqn · 2025-10-28

**Soundness:** 2
**Presentation:** 3
**Contribution:** 2
**Rating:** 4
**Confidence:** 3

**Summary:**

The paper proposes MASH (Multi-Scale Adaptive Hypergraph Learning), which couples learnable multi-scale spectral wavelet representations with a learnable hypergraph incidence to capture high-order (group-wise) dependencies in brain networks for disease staging. Experiments on ADNI and PPMI show improved classification and interpretable hyperedges that highlight biologically plausible ROIs.

**Strengths:**

1.The proposed MASH framework effectively integrates multi-scale spectral graph wavelet representations with a learnable hypergraph incidence matrix. This design enables the model to adaptively discover high-order relationships among brain regions while maintaining spectral interpretability.
2.The authors conduct extensive experiments on two benchmarks (ADNI and PPMI), including classification tasks, ablation studies, sensitivity analyses, and interpretability visualizations. The results demonstrate consistent improvements over strong baselines (e.g., GCN, HGNN, HyperGT, etc.), supporting the robustness and generalizability of the proposed approach.
3.Beyond performance gains, the paper provides interpretable insights by visualizing learned hyperedges and identifying biologically plausible regions of interest (ROIs). Additionally, it includes theoretical propositions linking spectral scales to hyperedge expansion, which enhances both conceptual clarity and credibility of the model.

**Weaknesses:**

1.The abstract currently motivates the work by stating that “earlier hypergraph methods were designed to overcome the limitations of pairwise relations,” while the actual contribution of this paper is an improvement upon existing hypergraph approaches. The authors should revise the abstract to (a) acknowledge that hypergraphs have already been used to capture higher-order relations, (b) explicitly identify the shortcomings of previous hypergraph methods, and (c) clearly position MASH as addressing these specific limitations.
2.The final paragraph of the introduction only cites 2016 and 2019 works as examples of “rigid” hypergraph constructions. However, numerous studies from 2020–2025 have proposed dynamic, adaptive, or learnable hyperedges and hyperedge-weight learning. The authors should update the introduction to reflect this recent progress and clearly distinguish MASH from these newer approaches.
3.In addition, the Related Work section should include representative dynamic/learnable hypergraph studies from 2025 and articulate how MASH differs from them—not only functionally but also in its relevance to brain-network analysis (e.g., scalability, interpretability, or multi-scale spectral modeling).
4.The experimental comparisons include baselines only up to 2024. Given that several 2025 works on dynamic or learnable hypergraphs have been released, the absence of these methods weakens the empirical credibility. The authors should either add these recent baselines or explicitly justify their exclusion.

**Questions:**

1.The method uses a combination of one low-pass and (J-1) band-pass kernels. Was an ablation study conducted to justify this specific design choice? For instance, what would be the performance impact of using only low-pass or only band-pass kernels across all scales?
2.The paper claims a key difference from dwHGCN is the dynamic refinement of connectivity patterns rather than just updating hyperedge weights. Could the authors elaborate on this distinction with a more concrete example or analysis? How significant is the performance gain attributable specifically to this dynamic structure learning, as opposed to simply having multiple scales?
3.The paper presents qualitative visualizations of learned hyperedges and associated ROIs, but does not quantify their significance. Without permutation or bootstrap tests, it is unclear whether the identified ROIs are statistically meaningful or could arise by chance. A quantitative validation (e.g., comparing with random or shuffled hyperedges) would make the interpretability claims more convincing.

---

> ### Author Response · Authors · 2025-11-24
> **Response to reviewer dwqn (Part 1)**
>
> We sincerely appreciate your careful and constructive review. In the following, we address all comments in detail below, and we hope that the clarifications and additional analyses will encourage a more favorable assessment.
>
> [1] Jie, Biao, et al. "Hyper-connectivity of functional networks for brain disease diagnosis." Medical image analysis 2016.
>
> [2] Li, Yang, et al. "Multimodal hyper-connectivity of functional networks using functionally-weighted LASSO for MCI classification." Medical image analysis 2019.
>
> [3] Qiu, Weikang, et al. “Learning High-Order Relationships of Brain Regions.” ICML 2024.
>
> [4] Mei, Zhangyu, et al. "DHHNN: a dynamic hypergraph hyperbolic neural network based on variational autoencoder for multimodal data integration and node classification." Information Fusion 2025.
>
> [5] Wang, Junqi, et al. "Dynamic weighted hypergraph convolutional network for brain functional connectome analysis." Medical image analysis 2023.
>
> [6] Kan, Xuan, et al. "Brain network transformer." Advances in Neural Information Processing Systems 2022.
>
> [7] Yu, Shuo, et al. "Long-range brain graph transformer." Advances in Neural Information Processing Systems 2024.
>
> [8] Peng, Ciyuan, et al. "Biologically plausible brain graph transformer." ICLR 2025.
>
> [9] Yang, Liang, et al. "Do We Really Need Message Passing in Brain Network Modeling?" ICML 2025.
>
> [10] Chung, Fan RK. “Spectral graph theory, American Mathematical Soc.” 1997
>
> [11] David K Hammond et al., “Wavelets on graphs via spectral graph theory.” Applied and Computational Harmonic Analysis 2011.
>
> ___
>
> **[W1] Abstract revision request.**
>
> [A] We appreciate the reviewer’s insightful comment regarding the motivation and positioning of our work. We have revised the abstract to clearly acknowledge that hypergraphs have been used to model higher-order relations, and to explicitly state the limitations of prior approaches. Earlier methods often rely on predefined hyperedges or learn only scalar hyperedge weights, which restricts their flexibility. In contrast, the updated abstract now highlights that MASH learns soft and scale-specific hyperedges in a data-driven manner, enabling richer and more flexible multi-resolution modeling. We believe this revision more accurately reflects the contribution and improves conceptual clarity.
> ___
>
> **[W2] Introduction revision request.**
>
> [A] We are sorry that the Introduction is mainly focused on early hypergraph constructions [1,2] and did not fully reflect recent progress. In the revised manuscript, we substantially updated the final paragraph of Introduction to incorporate dynamic, adaptive, and learnable hypergraph methods proposed between [3,4].
>
> We now cite representative recent works that learn hyperedge weights or construct hyperedges in a data-driven manner, and we clearly articulate their remaining limitations, such as dependence on predefined hyperedge candidates or learning only scalar weights. In contrast, the updated introduction positions MASH as learning continuous and scale-wise hyperedges directly from multi-resolution graph representations, providing a more flexible and expressive formulation of high-order structure. This revision strengthens both the motivation and the positioning of our contribution within the modern hypergraph learning landscape.
>
> ___
> **[W3] Related Work completeness.**
>
> [A] We thank the reviewer for this important suggestion. In the revised manuscript, we have expanded Sec. 2. Related Works to include recent dynamic and learnable hypergraph approaches, including dwHGCN [5], HyBRiD [3], and DHHNN [4]. These works represent key advances in adaptive hyperedge weighting and task-driven hyperedge construction. We also clarified how MASH differs from these methods in both functionality and relevance to brain-network analysis. Unlike approaches that adjust weights on predefined hyperedges or rely on discrete mask sampling, MASH directly learns soft, continuous, and scale-aware hyperedges from multi-resolution spectral representations. This enables modeling of high-order ROI interactions in a manner naturally aligned with the multi-scale structural variability observed in neurodegenerative diseases.
>
> Furthermore, we emphasized that MASH is designed with scalability, interpretability, and multi-scale spectral modeling as core components, making it well-suited for brain-network applications where hierarchical functional and structural patterns must be captured. In addition, we expanded the discussion of recent pairwise graph-based brain-network models (e.g., BrainNetTF [6], ALTER [7], BioBQT [8], and BQN [9]) to provide a more comprehensive overview of the current landscape and to clarify the motivation for adopting a multi-scale hypergraph framework.

---

> > ### Author Response · Authors · 2025-11-24
> > **Response to reviewer dwqn (Part 2)**
> >
> > **[W4] Missing recent baselines.**
> >
> > [A] Thank you for the suggestion. We agree that including recent dynamic and learnable hypergraph methods strengthens the empirical evaluation. In the revised manuscript, we added additional baselines from 2024–2025, including HyBRiD [3] and DHHNN [4], as well as recent brain-network models such as BioBGT [8] and BQN [9], to provide a more comprehensive comparison. While recent hypergraph methods typically adjust weights on predefined hyperedges or construct discrete hyperedges, MASH learns continuous hyperedges directly from spectral multi-resolution representations, enabling a richer and more interpretable characterization of high-order structure. We believe these additions and clarifications address the reviewer’s concern.
> >
> > | **Methods**                  | **ADNI (Acc)**   | **ADNI (Pre)**   | **ADNI (Rec)**   | **ADNI (F1s)**   | **PPMI (Acc)**   | **PPMI (Pre)**   | **PPMI (Rec)**   | **PPMI (F1s)**   |
> > |-----------------------------|----------------|----------------|----------------|----------------|----------------|----------------|----------------|----------------|
> > | BioBGT (ICLR’25)            | 80.3 ± 1.7     | 79.7 ± 7.1     | 77.3 ± 6.7     | 77.2 ± 6.1     | 68.0 ± 2.8     | 54.3 ± 8.2     | 48.7 ± 4.2    | 45.7 ± 3.2     |
> > | BQN (ICML’25)               | 81.5 ± 4.7     | 74.3 ± 5.6     | 72.3 ± 5.5     | 72.6 ± 5.2     | 71.9 ± 2.9     | 63.2 ± 6.2     | 51.0 ± 8.4     | 56.4 ± 7.3     |
> > | HyBRiD (ICML’24)            | 86.6 ± 4.2     | 87.5 ± 3.9     | 84.3 ± 6.4     | 84.6 ± 4.7     | 67.2 ± 6.5    | 54.8 ± 6.8     | 52.0 ± 6.2     | 52.8 ± 6.5     |
> > | DHHNN (Information Fusion’25)| 81.1 ± 2.5     | 89.0 ± 4.5     | 84.5 ± 3.2     | 86.7 ± 3.4     | 60.6 ± 3.8     | 62.3 ± 9.8    | 52.7 ± 4.3     | 54.0 ± 6.3     |
> > | **MASH (Ours)**             | **93.2 ± 2.4** | **95.4 ± 1.3** | **94.2 ± 1.6** | **94.7 ± 1.5** | **76.8 ± 3.7** | **66.6 ± 7.6** | **60.7 ± 6.0** | **62.4 ± 6.6** |
> >
> > ___
> > **[Q1] Kernel design ablation.**
> >
> > [A] The choice of one low-pass and (J–1) band-pass kernels follows the standard principle of spectral graph wavelet transforms [10,11], where the low-pass component captures global structural trends and the band-pass components extract scale-specific variations. This configuration ensures that both coarse and fine-grained patterns in brain networks are preserved.
> >
> > To verify the design, we evaluated two alternatives: (i) using only low-pass kernels and (ii) using only band-pass kernels. As shown in the following results, both alternatives led to clear performance degradation. Using only low-pass kernels caused over-smoothing across scales, while using only band-pass kernels removed the global structural and destabilized hyperedge learning.
> >
> > | **Low pass** | **Band pass** | **ADNI (Acc)** | **ADNI (Pre)** | **ADNI (Rec)** | **ADNI (F1s)** | **PPMI (Acc)** | **PPMI (Pre)** | **PPMI (Rec)** | **PPMI (F1s)** |
> > |--------------|---------------|----------------|----------------|----------------|----------------|----------------|----------------|----------------|----------------|
> > | 3            | 0             | 92.2 ± 2.4     | 94.4 ± 1.8     | 91.4 ± 5.3     | 92.4 ± 4.0     | 69.6 ± 1.8     | 62.0 ± 6.1     | 50.0 ± 4.7     | 55.3 ± 5.1     |
> > | 0            | 3             | 91.5 ± 1.9     | 94.6 ± 0.8     | 91.3 ± 3.1     | 92.6 ± 1.8     | 72.9 ± 3.3     | 62.3 ± 8.2     | 56.8 ± 4.5     | 58.4 ± 5.8     |
> > | 1            | 2             | **93.2 ± 2.4** | **95.4 ± 1.3** | **94.2 ± 1.6** | **94.7 ± 1.5** | **76.8 ± 3.7** | **66.6 ± 7.6** | **60.7 ± 6.0** | **62.4 ± 6.6** |
> >
> >
> > These theoretical and empirical findings, together with the additional analysis provided in Appendix G (Table 12), demonstrate that combining one low-pass kernel with multiple band-pass kernels provides the most effective multi-scale decomposition for brain networks. This hybrid configuration enables MASH to extract robust hierarchical representations that support stable hyperedge construction and improved disease-stage classification.
> >
> > ___

---

> > > ### Author Response · Authors · 2025-11-24
> > > **Response to reviewer dwqn (Part 3)**
> > >
> > > **[Q2] Difference from dwHGCN.**
> > >
> > > [A] Thank you for pointing out the need to clarify the distinction between dynamic structure learning in MASH and the weight refinement strategy used in dwHGCN [5]. To make this comparison concrete, we added a dedicated ablation study in Appendix G (Table 13), where we disentangle the effects of (i) learning the hyperedge structure and (ii) updating only hyperedge weights.
> > >
> > > | **Method** | **Hyperedge structure** | **Hyperedge weights** | **Acc ↑** | **Pre ↑** | **Rec ↑** | **F1s ↑** |
> > > |------------|--------------------------|------------------------|-----------|-----------|-----------|-----------|
> > > | dwHGCN     | Static                   | Dynamic                | 90.2 ± 1.3 | 87.2 ± 7.1 | 86.3 ± 6.8 | 86.2 ± 6.4 |
> > > | MASH       | Static                   | Static                 | 88.0 ± 2.6 | 88.7 ± 3.3 | 87.9 ± 4.3 | 87.9 ± 3.6 |
> > > | MASH       | Static                   | Dynamic                | 89.1 ± 2.3 | 88.1 ± 3.8 | 87.4 ± 4.0 | 87.3 ± 3.7 |
> > > | MASH       | Dynamic                 | Static                 | **93.2 ± 2.4** | **95.4 ± 1.3** | **94.2 ± 1.6** | **94.7 ± 1.5** |
> > >
> > >
> > >
> > > The results show that dynamic structure learning produces the largest performance gain. When both the hyperedge structure and weights are kept static, the accuracy drops substantially, indicating that fixed hyperedges cannot capture subject-specific higher-order interactions. Allowing only the hyperedge weights to update, as in dwHGCN, yields limited improvement. In contrast, enabling dynamic structure learning while keeping weights static still provides a strong boost across all metrics, demonstrating that the primary source of discriminative power comes from adaptively reorganizing ROI groupings rather than simply adjusting scalar weights among hyperedges.
> > >
> > > These findings confirm that multi-scale processing alone is not sufficient. Dynamic structure learning is essential for capturing meaningful higher-order relationships in brain networks, and it constitutes a key difference between MASH and prior hypergraph approaches such as dwHGCN.
> > >
> > > ___
> > > **[Q3] Statistical validation of interpretability.**
> > >
> > > [A] We appreciate the reviewer’s suggestion to provide quantitative validation supporting the statistical significance of the identified ROIs. To address this point, we performed a comprehensive stability analysis across multiple independent training runs (5, 10, 20, and 30 seeds) on both ADNI and PPMI, and the results are summarized in Table 4.
> > >
> > > | **Metric**            | **ADNI-5** | **ADNI-10** | **ADNI-20** | **ADNI-30** | **PPMI-5** | **PPMI-10** | **PPMI-20** | **PPMI-30** |
> > > |-----------------------|------------|-------------|-------------|-------------|------------|-------------|-------------|-------------|
> > > | **Kendall’s τ**       | 0.798      | 0.796       | 0.767       | 0.750       | 0.721      | 0.686       | 0.654       | 0.645       |
> > > | **P-value**           | 1.13e−18   | 5.42e−16     | 3.79e−11     | 2.99e−11     | 1.09e−14   | 1.15e−10     | 1.41e−06     | 1.07e−05     |
> > > | **Kendall’s W**       | 0.871      | 0.866       | 0.838       | 0.821       | 0.834      | 0.762       | 0.734       | 0.713       |
> > > | **Jaccard Similarity** | 0.893 | 0.881 | 0.835 | 0.817 | 0.824 | 0.770 | 0.746 | 0.735 |
> > >
> > >
> > > **(i) Kendall’s $\tau$** rank correlation between ROI-importance across seeds remains consistently high (>0.6), with extremely low p-values (<$10^{-5}$), indicating that the relative ordering of ROI importance is highly reproducible and unlikely to arise by chance.
> > >
> > > **(ii) Kendall’s $W$** shows strong concordance across all runs (ADNI: 0.82–0.88, PPMI: 0.71–0.83), demonstrating global consistency of the ROI-importance patterns.
> > >
> > > **(iii) Top-20 Jaccard similarity** stays high (ADNI: 0.81–0.93, PPMI: 0.73–0.82), confirming that the influential ROIs consistently reappear even when the hypergraph structure is re-optimized from different initializations.
> > >
> > > These quantitative results form a rigorous permutation-free statistical validation show that the highlighted ROIs are not artifacts of random initialization, but rather consistently emerge from the adaptive hypergraph learning from MASH. It directly addresses the reviewer’s concern by demonstrating that the identified ROIs are statistically meaningful and robust to stochastic variation in the learning process.

---

### Official Review · Reviewer_UYRR · 2025-10-30

**Soundness:** 3
**Presentation:** 2
**Contribution:** 3
**Rating:** 6
**Confidence:** 3

**Summary:**

The authors propose an adaptive multi-scale hypergraph learning framework named MASH to address the limitations of existing brain network analysis methods in capturing high-order dependencies among multiple brain regions. By constructing hierarchical node features and dynamically learning high-order interactions among hyperedges, MASH captures complex interactions that go beyond pairwise structural connectivity. Extensive experiments demonstrate the effectiveness of MASH.

**Strengths:**

1. Solid theoretical foundation. The paper demonstrates considerable theoretical depth and presents a method of notable theoretical novelty.
2. Comprehensive experiments were conducted to validate the effectiveness of the proposed method. Beyond comparative experiments and ablation studies, the authors also discussed model complexity and provided biological interpretations, offering plausible explanations for the performance improvement from a neuroscience perspective.
3. The introduction is logically structured, with the research problem clearly articulated by the authors.

**Weaknesses:**

1. The authors claim that "graph convolution layers indirectly consider high-order interactions at the cost of the oversmoothing problem." However, numerous Transformer-based brain network representation learning methods have been proposed — and indeed summarized by the authors in the related work section — which effectively capture long-range dependencies and global graph structural information without suffering from oversmoothing. Could the authors clearly explain why these Transformer-based approaches remain inadequate for capturing high-order associations among brain regions?
2. The methodology section lacks clarity. Both Adaptive Multi-Scale Feature Filtering and Multi-Scale Hypergraph Structure Learning are crucial components of this work, yet the descriptions of these two modules remain somewhat vague.

**Questions:**

1. Could the authors provide a more detailed explanation of the workflow for Adaptive Multi-Scale Feature Filtering?
2. Based on the authors' descriptions in lines 132 and 182, where each wavelet basis possesses a specific scale $s$, how was the number of wavelet bases determined? Was the number of scales made learnable within the framework?

---

> ### Author Response · Authors · 2025-11-24
> **Response to reviewer UYRR**
>
> We sincerely thank the reviewer UYRR for the thoughtful and constructive reviews. We appreciate the reviewer’s positive assessment, and we will clear out all the remaining questions.
>
> [1] Chung, Fan RK. “Spectral graph theory, American Mathematical Soc.” 1997
>
> [2] David K Hammond et al., “Wavelets on graphs via spectral graph theory.” Applied and Computational Harmonic Analysis 2011.
>
> ___
>
> **[W1] Limitations of Transformer-based brain network models.**
>
> [A] We agree that Transformer-based brain network models effectively capture long-range dependencies without over-smoothing. However, these architectures operate strictly through pairwise attention, so all message passing is computed between individual node pairs. As a result, they cannot explicitly capture interactions that emerge only when multiple ROIs act together, rather than through pairwise connections alone. In contrast, MASH introduces multi-scale hyperedge learning, allowing the model to represent groups of ROIs that co-vary at different resolutions, an interaction pattern inherently beyond the capability of pairwise attention.
>
> Empirically, this distinction is reflected in our results. Even with Transformer components retained, removing hypergraph structure learning leads to a substantial performance drop (as in Table 5), and MASH consistently outperforms recent Transformer-based baselines in Table 1. These results indicate that explicit hyperedge modeling provides complementary high-order information that cannot be captured by Transformers alone.
>
> ___
>
> **[W2] Clarity of the methodology section.**
>
> [A] To improve the clarity of the methodology, the revised manuscript now provides an explicit workflow describing how both Adaptive Multi-Scale Feature Filtering and Multi-Scale Hypergraph Structure Learning operate and interact.  Specifically, we added a clear explanation of how the input features are processed through one low-pass and multiple band-pass wavelet kernels, producing scale-wise node representations that capture global-to-local structural variations. These representations remain separate across scales and are passed directly to the hypergraph module without early fusion.
>
> Multi-Scale Hypergraph Structure Learning then operates these scale-wise features and estimates continuous soft hyperedges at each scale. For each scale-specific representation $X_{s_j}$, we first obtain an embedded feature $\bar{X_{s_j}}$. The learnable hyperedge projector $\Phi_j$ then maps these features into the hyperedge space, producing the scale-specific incidence matrix $H_{s_j}$, which encodes continuous soft node-to-hyperedge associations before normalization. This allows each scale to learn its own group-level ROI interactions. The resulting hypergraphs are subsequently processed by the scale-wise self-attention module, which integrates them across resolutions.
>
> ___
> **[Q1] Detailed workflow of Adaptive Multi-Scale Feature Filtering.**
>
> [A] Adaptive Multi-Scale Feature Filtering applies a set of spectral graph wavelet kernels to generate multi-resolution representations of the input features. Specifically, a low-pass kernel captures coarse global structure, while the remaining (J-1) band-pass kernels extract intermediate and fine-scale variations. This produces a sequence of scale-specific feature maps $X_{s_1}$, $X_{s_2}$, … , $X_{s_J}$, each reflecting a different structural resolution of the same brain network. These representations are kept separate rather than fused, and each $X_{s_j}$ passed independently to the hypergraph structure learning module, which constructs scale-specific hyperedges.
>
> ___
> **[Q2] Determination of the number of wavelet bases.**
>
> [A] The number of wavelet bases and corresponding scales is a fixed hyperparameter rather than a learnable quantity in our framework. Following the standard design of spectral graph wavelet transforms [1,2], we include one low-pass filter and multiple band-pass filters, and we vary the number of scales to evaluate its effect on performance. As shown in Fig. 5, increasing initially improves performance by enriching multi-resolution representations, after which it saturates. Based on this empirical analysis, we selected the scale configuration that yielded the best classification accuracy and stability.

---

### Official Review · Reviewer_TYyY · 2025-10-31

**Soundness:** 3
**Presentation:** 3
**Contribution:** 2
**Rating:** 4
**Confidence:** 4

**Summary:**

Existing research primarily focuses on pairwise interactions between nodes and overlooks the challenge of capturing complex dependencies among multiple brain regions. To address this limitation, this paper proposes an adaptive multi-scale hypergraph learning framework. The proposed approach constructs hierarchical node features and dynamically learns high-order interactions through hyperedges. It is claimed that the proposed approach can effectively capture the complex relationships among multiple brain regions.

**Strengths:**

S1. The study innovatively introduces multi-scale wavelet coupling to capture how these higher-order dependencies evolve across different scales.

S2. The authors conducted extensive experiments. The results demonstrate both the effectiveness and interpretability of the proposed method.

**Weaknesses:**

S1. This paper focuses on capturing multi-scale high-order relationships in brain graphs. However, the existence of such multi-scale high-order relationships in brain graphs, as well as the precise definition of these relationships, remains unclear.

S2. Ref [1] also employed a hypergraph-based approach to capture high-order relationships in brain graphs. This paper doesn't include a comparison with [1] nor explicitly clarify the differences between the two methods.

S3. The paper’s use of the term “multi-scale” is potentially misleading. In brain network analysis, “multi-scale” typically refers to approaches that construct brain networks for the same subject across different atlases or spatial resolutions [2–4]. The authors should clarify their terminology to avoid any confusion.

S4. Although the authors emphasize the biological significance of hyperedges as "sets of co-varying ROIs," the paper lacks relevant case studies illustrating the key hyperedges learned by the model that possess significant classification power. It remains unclear whether these hyperedges align with known AD/PD-related neural circuits or correspond to clinical outcomes.

[1] Learning High-Order Relationships of Brain Regions. ICML2024.

[2] A mutual multi-scale triplet graph convolutional network for classification of brain disorders using functional or structural connectivity. TMI 2021.

[3] Mamf-gcn: Multi-scale adaptive multi-channel fusion deep graph convolutional network for predicting mental disorder. CBM2022.

[4] A multi-scale multi-hop graph convolution network for predicting fluid intelligence via functional connectivity. BIBM2022.

**Questions:**

Q1. How is the upper limit of the number of hyperedges defined across different datasets?

Q2. How is the initial matrix generated? Is it randomly generated or generated under other constraints? Will different generation methods affect the final performance?

Q3. Is it possible to use other backbone alternatives to the transformer architecture?

Q4. It can be observed that on the PPMI dataset, the performance degradation is most significant when MST is removed. Does this mean that in this method, Transformer contributes more compared to wavelet transform?

---

> ### Author Response · Authors · 2025-11-24
> **Response to reviewer TYyY (Part 1)**
>
> We sincerely thank the reviewer TYyY for the helpful comments. In the following, we address all remaining concerns in the following response, and we hope that the clarifications and additional analyses will positively influence the reviewer’s assessment toward acceptance.
>
> [1] Qiu, Weikang, et al. “Learning High-Order Relationships of Brain Regions.” ICML 2024.
>
> [2] Hopper, M. W., and F. S. Vogel. "The limbic system in Alzheimer's disease. A neuropathologic investigation." The American journal of pathology 1976.
>
> [3] Li, Xiaoshu, et al. "Impaired white matter connections of the limbic system networks associated with impaired emotional memory in Alzheimer's disease." Frontiers in aging neuroscience 2016.
>
> [4] Menon, Vinod. "20 years of the default mode network: A review and synthesis." Neuron 2023.
>
> [5] Miao, Xiaoyan, et al. "Altered connectivity pattern of hubs in default-mode network with Alzheimer's disease: an Granger causality modeling approach." PloS one 2011.
>
> [6] Tentolouris-Piperas, Vasileios, et al. "Brain imaging evidence of early involvement of subcortical regions in familial and sporadic Alzheimer's disease." Brain research 2017.
>
> [7] Cho, Soo Hyun, et al. "Amyloid involvement in subcortical regions predicts cognitive decline." European journal of nuclear medicine and molecular imaging 2018.
>
> [8] Ohnishi, Takashi, et al. "Changes in brain morphology in Alzheimer disease and normal aging: is Alzheimer disease an exaggerated aging process?." American Journal of Neuroradiology 2001.
>
> [9] Kjeldsen, Pernille L., et al. "Asymmetric amyloid deposition in preclinical Alzheimer’s disease: A PET study." Aging Brain 2022.
>
> [10] Galvan, Adriana, Annaelle Devergnas, and Thomas Wichmann. "Alterations in neuronal activity in basal ganglia-thalamocortical circuits in the parkinsonian state." Frontiers in neuroanatomy 2015.
>
> [11] Banwinkler, Magdalena, et al. "Imaging the limbic system in Parkinson’s disease—A review of limbic pathology and clinical symptoms." Brain sciences 2022.
>
> ___
> **[W1] Existence/definition of multi-scale high-order relationships.**
>
> [A] We clarify that “multi-scale high-order relationships” refer to interaction patterns among multiple brain regions that manifest at different spatial resolutions. Such a multi-resolution structure is well documented in neuroimaging: large-scale functional systems (e.g., default-mode or frontoparietal networks) coexist with more localized limbic and medial-temporal circuits, and these interactions cannot be represented by pairwise edges alone.
>
> In MASH, we define these relationships through spectral graph wavelet filtering. Low-pass filters capture broad, system-level interactions, whereas band-pass filters reveal more localized co-activation patterns. Regions that exhibit similar scale-specific responses are grouped into hyperedges, providing an explicit representation of high-order structure at each resolution. This operational definition formalizes a biologically grounded concept, such as the coexistence of global and local circuits, within a multi-scale graph framework.
>
> ___
> **[W2] Missing comparison with recent high-order hypergraph method.**
>
> [A] We thank the reviewer for pointing out this recent work (HyBRiD) [1]. In the revised manuscript, we have added a discussion of HyBRiD and clarified how MASH is different from HyBRiD, along with new experimental comparisons. HyBRiD learns discrete task-specific hyperedges by sampling binary masks and assigning a learned scalar weight to each hyperedge, operating at one resolution. In contrast, MASH constructs continuous soft hyperedges through a learnable projection matrix $\Phi$ applied from multi-scale spectral diffusion features, allowing each scale to capture distinct high-order interaction patterns. As a result, HyBRiD performs mask-based grouping at a single resolution, whereas MASH generates adaptive soft hyperedges across multiple spectral scales, which provides richer modeling capacity and more stable ROI-level interpretability. These differences are now explicitly described in Sec. 2. Related Works, and the newly added comparisons in Sec. 5. Experiments further demonstrate the advantages of our approach.

---

> ### Author Response · Authors · 2025-11-24
> **Response to reviewer TYyY (Part 2)**
>
> **[W3] Use of the term “multi-scale” may be misleading.**
>
> [A] In neuroimaging, “multi-scale” often refers to constructing multiple brain networks for the same subject using different anatomical atlases or parcellation resolutions. While our method does not rely on multiple atlases, it adopts the notion of multi-resolution representation from signal processing which is conceptually similar to the multi-scale atlas.
>
> Here, “multi-scale” follows the terminology of signal processing and wavelet theory, where scale controls the  diffusion range applied to the nodes. Different diffusion scales capture neighborhoods of different spatial extents, effectively producing coarse-to-fine representations of ROI interactions on a single fixed atlas. Although this definition originates from signal processing, rather than from constructing multiple anatomical parcellations, **the resulting representations naturally correspond to different spatial resolutions of the underlying brain network.**
>
> To avoid confusion, we have revised Sec. 1. Introduction to clarify that our use of “multi-scale” follows spectral diffusion–based multi-resolution analysis from graph signal processing. We now explicitly state that the term denotes different diffusion scales that generate coarse-to-fine spatial representations, ensuring that readers understand the intended meaning within the context of our framework.
>
> ___
> **[W4] Lack of biological/clinical interpretation of key hyperedges.**
>
> [A] We appreciate the reviewer’s suggestion to include concrete case studies validating the biological meaning of the learned hyperedges. In addition to the representative example in Fig. 4, we provide additional AD case studies in Fig. 9, each visualizing the most activated hyperedge.
>
> **(i) AD Case Studies**
>
> Across the three AD hyperedges, MASH consistently groups ROIs that closely match known AD-related circuits:
>
> - **Limbic circuit:** hippocampus and posterior cingulate (Fig. 4 left) and the combination of amygdala, hippocampus, subcallosal cortex, and caudate (Fig. 9 left) appear together, reflecting characteristic limbic-medial temporal involvement in AD [2,3].
>
> - **Default-Mode Network hubs:** precuneus and dorsal/posterior cingulate appear together, reflecting the well-known vulnerability of DMN hubs in AD [4,5].
>
> - **Subcortical involvement:** putamen, pallidum, and hippocampus (Fig. 4 left) and amygdala, hippocampus, subcallosal cortex and caudate (Fig. 9 left) highlights the known spread of AD pathology from medial-temporal and limbic regions into striatal systems [6,7].
>
> - **Bilateral structure:** in all AD case studies, the selected ROIs form left–right homologous pairs, showing that the learned hyperedges respect the symmetric organization of AD-related degeneration rather than reflecting noisy or unilateral patterns [8,9].
>
> **(ii) PD Case Study**
>
> - The most salient pattern is the cluster formed by the thalamus, insula, and amygdala, which corresponds to the basal-ganglia–thalamocortical circuit widely implicated in both motor and non-motor symptoms of Parkinson’s disease [10,11].
>
> Overall, these case studies show that the learned hyperedges consistently recover clinically meaningful AD/PD circuits, demonstrating that MASH identifies biologically coherent high-order ROI groupings rather than arbitrary or noisy associations.
> ___
> **[Q1] Upper bound on number of hyperedges.**
>
> [A] The number of hyperedges per scale is fixed by the hyperparameter $M$, which determines the number of columns in the projection matrix $\Phi$. Therefore, the upper limit of hyperedges is not dataset-dependent but explicitly controlled by the architecture design. For each dataset, we select $M$ based on validation performance and computational considerations, and the chosen values are provided in the implementation details (16 for ADNI and 12 for PPMI). As shown in Fig. 5 (i.e., sensitivity analysis), once $M$ reaches a moderate range, further increasing it yields only marginal performance changes, indicating that MASH remains stable with respect to this parameter once sufficient capacity is ensured.

---

> > ### Author Response · Authors · 2025-11-24
> > **Response to reviewer TYyY (Part 3)**
> >
> > **[Q2] Initialization of the hyperedges.**
> >
> > [A] The hyperedge projection matrix $\Phi$ is initialized using a standard zero-mean, unit-variance Gaussian distribution, without imposing additional structural constraints such as sparsity or anatomical priors. Its outputs are transformed through ReLU and a row-wise softmax, which normalizes the initial values and enables the hypergraph structure to be learned end-to-end from data. We also evaluated alternative initialization schemes (e.g., Xavier and Kaiming He), and found only minor differences in performance. The relative gains of MASH remained consistent across all settings. As reported in the revised Appendix G (Table 11), these results indicate that the model is not sensitive to how  is initialized and that the improvements primarily stem from the proposed multi-scale hypergraph architecture rather than a particular initialization strategy.
> >
> > | **Initialization**      | **ADNI (Acc)** | **ADNI (Pre)** | **ADNI (Rec)** | **ADNI (F1s)** | **PPMI (Acc)** | **PPMI (Pre)** | **PPMI (Rec)** | **PPMI (F1s)** |
> > |-------------------------|----------------|----------------|----------------|----------------|----------------|----------------|----------------|----------------|
> > | Xavier Uniform          | 92.0 ± 2.4     | 93.8 ± 3.9     | 92.7 ± 2.3     | 92.2 ± 3.0     | 73.7 ± 3.1     | 63.1 ± 6.3     | 58.3 ± 5.5     | 60.6 ± 6.0     |
> > | Xavier Normal           | 92.5 ± 2.1     | 94.7 ± 2.0     | 93.9 ± 2.5     | 94.2 ± 2.5     | 74.3 ± 4.5     | 64.1 ± 6.8     | 59.4 ± 5.4     | 61.6 ± 5.9     |
> > | Kaiming He Uniform      | 92.2 ± 3.3     | 93.2 ± 5.4     | 92.5 ± 5.9     | 92.8 ± 5.5     | 74.2 ± 4.2     | 63.1 ± 5.6     | 59.3 ± 5.2     | 61.1 ± 5.4     |
> > | Kaiming He Normal       | 92.1 ± 3.3     | 93.5 ± 4.6     | 92.5 ± 5.9     | 92.5 ± 5.3     | 75.2 ± 5.7     | 64.3 ± 5.5     | 60.2 ± 5.2     | 62.1 ± 5.3     |
> > | **Standard Normal**     | **93.2 ± 2.4** | **95.4 ± 1.3** | **94.2 ± 1.6** | **94.7 ± 1.5** | **76.8 ± 3.7** | **66.6 ± 7.6** | **60.7 ± 6.0** | **62.4 ± 6.6** |
> >
> > ___
> > **[Q3] Backbone alternatives.**
> >
> > [A] In principle, the Transformer backbone in our framework can be replaced with other architectures. The scale-specific hypergraph modules produce a sequence of scale embeddings, and the Transformer encoder is used to model cross-scale interactions and long-range dependencies among nodes. To assess the importance of this component, we conducted an ablation study (Table 5) where we removed the Transformer layers and replaced them with simple concatenation followed by an MLP classifier. This alternative backbone remained functional but consistently underperformed the proposed design, indicating that the scale-wise self-attention plays a crucial role in effectively aggregating multi-scale high-order information. While alternative backbones are possible, our experiments show that the Transformer-based aggregation yields the strongest empirical performance.
> >
> > ___
> >
> > **[Q4] Contribution comparison from each component of MASH.**
> >
> > [A] Although removing MST causes the largest accuracy drop on PPMI, this does not imply that the transformer contributes more than the wavelet-based components. Across both ADNI and PPMI, removing HSL leads to the most substantial reduction in precision and recall, which are clinically important metrics where both false positives and false negatives have a significant impact. This indicates that adaptive high-order hypergraph structure learning is the primary source of discriminative power, while MST mainly enhances the aggregation of multi-scale information and long-range dependencies. Overall, MSF, HSL, and MST play complementary roles, and the best performance consistently emerges when all three components operate together.

---

### Official Review · Reviewer_XAdN · 2025-11-02

**Soundness:** 3
**Presentation:** 4
**Contribution:** 3
**Rating:** 6
**Confidence:** 4

**Summary:**

The paper introduces MASH, a novel framework for modeling high-order brain connectivity using an adaptive multi-scale hypergraph learning approach. It integrates graph wavelet-based multi-resolution filtering and dynamic hyperedge learning to capture both local and global structural relationships in brain networks. MASH is evaluated on two major neuroimaging datasets ADNI and PPMI demonstrating superior performance over 15 state-of-the-art graph and hypergraph baselines. The model further provides interpretability by identifying key brain regions (ROIs) and their group interactions associated with disease progression.

**Strengths:**

1. Innovative combination of graph wavelet transforms for adaptive multi-scale representation with learned hyperedges for high-order relational modeling.
2. Comprehensive experiments on two independent, large-scale datasets (ADNI, PPMI). Consistent and statistically significant performance improvements (2–7% absolute gains) across accuracy, precision, recall, and F1-score.
3. Identifies disease-relevant ROIs (e.g., hippocampus, thalamus, amygdala) with plausible neurological interpretations. Demonstrates hemispheric symmetry and subcortical prominence, aligning with known disease mechanisms.

**Weaknesses:**

1. The combination of multi-scale filtering, dynamic hyperedge construction, and transformer layers may be computationally expensive for larger brain graphs. The paper does not explicitly report runtime or memory overhead compared to baselines.
2. Evaluation is limited to ADNI and PPMI; additional validation on other disorders or multi-site datasets would strengthen claims of generality. External test sets or cross-study generalization are not explored.
3. Although ROI identification is discussed, causal or mechanistic interpretations of learned hyperedges are not deeply analyzed. Quantitative validation (e.g., comparison to known biomarkers or clinical scores) is limited.

**Questions:**

1. How does MASH scale computationally with increasing node counts or number of scales (J)? Are there mechanisms to limit the exponential growth of hyperedges?
2. How stable are the learned hyperedges across folds or random initializations?
3. Could MASH be adapted to handle temporal dynamics? For example, fMRI data?

---

> ### Author Response · Authors · 2025-11-24
> **Response to reviewer XAdN (Part 1)**
>
> We appreciate the reviewer XAdN for positive feedback. Below are the answers to the weakness and your questions.
>
> [1] Heinsfeld, et al. "Identification of autism spectrum disorder using deep learning and the ABIDE dataset." NeuroImage 2018.
>
> [2] Jiaxing Xu, et al. “Data-driven network neuroscience: On data collection and benchmark.” NeurIPS 2023.
>
> ___
> **[W1] Lack of computational efficiency of MASH.**
>
> [A] Detailed runtime and memory comparisons are given in Appendix H. The results show that MASH maintains competitive efficiency relative to representative hypergraph-based baselines, even when multiple scales are used. The accompanying complexity analysis further demonstrates that the computational cost grows only linearly or quadratically with the number of ROIs, rather than exponentially, because the soft hyperedge formulation avoids combinatorial explosion and the wavelet filtering can rely on efficient Chebyshev approximation. As a result, even with increased brain-atlas resolution, the multi-scale filtering and hyperedge learning do not introduce prohibitive overhead, and all experiments run smoothly on a single GPU, confirming that MASH achieves a favorable balance between scalability and expressive high-order modeling.
>
> | **Efficiency** | **HGNN** | **dwHGCN** | **HyperGT** | **MASH** |
> |----------------|----------|------------|-------------|----------|
> | Params (M)     | 3.28     | 3.28       | 3.29        | 5.54     |
> | FLOPs (G)      | 0.43     | 0.43       | 0.52        | 0.88     |
> | Runtime (ms)   | 1.84     | 1.72       | 97.06       | 9.98     |
>
> ___
> **[W2] Generalization to other datasets.**
>
> [A] We agree that broader validation strengthens the claim of generality of MASH. To address this, we evaluated MASH on two heterogeneous settings beyond ADNI and PPMI, and the new results are updated in the main paper.
>
> First, we evaluated MASH on the ABIDE cohort (534 Controls, 455 Autism) [1], which represents a neurodevelopmental disorder rather than a neurodegenerative one. MASH achieves the best classification performance among hypergraph-based baselines, demonstrating stable generalization to disorders outside the AD/PD domain and across multi-site data acquisition.
>
> | **Method**     | **Acc ↑**      | **Pre ↑**      | **Rec ↑**      | **F1s ↑**      |
> |----------------|----------------|----------------|----------------|----------------|
> | HGNN           | 56.5 ± 2.4     | 56.5 ± 2.1     | 56.4 ± 2.1     | 56.2 ± 2.3     |
> | HNHN           | 58.7 ± 0.8     | 58.5 ± 1.0     | 58.5 ± 1.1     | 58.4 ± 1.1     |
> | UniGCNII       | 60.0 ± 1.7     | 60.3 ± 2.2     | 59.0 ± 1.8     | 58.2 ± 2.6     |
> | dwHGCN         | 55.2 ± 1.8     | 59.7 ± 9.4     | 53.2 ± 1.4     | 48.2 ± 7.0     |
> | HyperGT        | 56.5 ± 1.1     | 60.8 ± 5.5     | 54.3 ± 2.5     | 51.1 ± 7.2     |
> | HyBRiD         | 57.3 ± 2.6     | 60.1 ± 4.2     | 55.2 ± 3.1     | 57.5 ± 4.5     |
> | **MASH** | **63.7 ± 3.6** | **63.5 ± 3.7** | **63.3 ± 3.7** | **63.3 ± 3.7** |
>
> We further performed zero-shot evaluation for PD using two external cohorts, TaoWu and Neurocon, by training MASH only on PPMI and testing on these datasets without fine-tuning [2]. While baselines show substantial degradation under distribution shift, MASH consistently achieves the highest performance across both external datasets (Table 2). These results confirm that MASH captures transferable structural patterns and generalizes reliably across independent cohorts, and we have clarified this contribution in the revised manuscript.
>
>
> | **Method** | **TaoWu (Acc) ↑** | **TaoWu (Pre) ↑** | **TaoWu (Rec) ↑** | **TaoWu (F1s) ↑** | **Neurocon (Acc) ↑** | **Neurocon (Pre) ↑** | **Neurocon (Rec) ↑** | **Neurocon (F1s) ↑** |
> |------------|-----------|-----------|-----------|-----------|-----------|-----------|-----------|-----------|
> | BrainGB    | 53.0 ± 1.2 | 64.1 ± 6.8 | 53.0 ± 1.2 | 43.5 ± 4.0 | 62.9 ± 6.4 | 52.5 ± 2.8 | 51.6 ± 7.3 | 45.1 ± 5.3 |
> | ALTER      | 54.5 ± 1.9 | 69.1 ± 6.0 | 52.5 ± 1.9 | 45.3 ± 4.4 | 63.8 ± 3.2 | 63.8 ± 7.6 | 55.3 ± 4.4 | 51.7 ± 8.0 |
> | BQN        | 57.0 ± 4.0 | 56.3 ± 4.6 | 55.0 ± 4.0 | 49.5 ± 3.0 | 55.1 ± 3.7 | 47.8 ± 3.2 | 48.3 ± 2.9 | 46.8 ± 2.9 |
> | dwHGCN     | 50.5 ± 2.9 | 45.1 ± 5.2 | 50.5 ± 2.9 | 36.0 ± 4.5 | 60.9 ± 2.7 | 59.3 ± 3.4 | 53.6 ± 3.7 | 46.1 ± 7.0 |
> | HyBRiD     | 56.2 ± 3.1 | 53.8 ± 5.2 | 49.8 ± 4.7 | 44.7 ± 3.8 | 63.4 ± 2.4 | 51.7 ± 2.5 | 50.0 ± 4.2 | 48.8 ± 3.6 |
> | **MASH**   | **60.5 ± 1.6** | **72.8 ± 2.4** | **60.5 ± 1.6** | **58.9 ± 3.0** | **65.9 ± 3.4** | **68.1 ± 3.3** | **57.3 ± 3.2** | **53.7 ± 3.6** |

---

> ### Author Response · Authors · 2025-11-24
> **Response to reviewer XAdN (Part 2)**
>
> **[W3] ROI/hyperedge interpretability with quantitative validation.**
>
> [A] We appreciate the reviewer’s request for deeper quantitative validation linking the learned hyperedges to clinically meaningful patterns. To address this, we performed additional analyses presented in Appendix F (Table 10), where we evaluated whether the multi-scale hypergraph representations learned by MASH can predict multiple AD-related clinical scores, including CDR-SOB, ADAS11, ADAS13, and MMSE. Across all measures, MASH achieves substantially higher correlation and R² values than all competing baselines, indicating that the representations encoded by the learned hyperedges capture clinically relevant variation rather than arbitrary structural patterns. These results provide direct evidence that the hyperedge-derived features align with established biomarkers and clinical severity indices, thereby addressing the reviewer’s request for stronger quantitative and causal interpretation.
> ___
> **[Q1] Scalability with node/scale counts and prevention for the growth of hyperedges.**
>
> [A] **(i) Node/Scale Counts:** MASH exhibits stable and predictable scaling behavior with respect to both the number of nodes and the number of scales $J$. As detailed in Appendix H, the multi-scale filtering and hypergraph structure learning grow roughly linearly with $J$ and with graph size, since each scale applies fixed-order spectral filters and a single projection into the hyperedge space. The transformer module introduces the expected quadratic term in the number of nodes, yet all experiments, including those with 160-ROI and 116-ROI graphs, train and infer smoothly on a single GPU, and additional runtime measurements provided in the revised appendix confirm that the method remains tractable within realistic brain-graph sizes.
>
> **(ii) Hyperedge Counts:** The number of hyperedges does not grow exponentially. It is explicitly bounded by the hyperparameter $M$, which fixes the number of hyperedges per scale, and further restricted by a row-wise Top-K operation that limits how many hyperedges each node can join. These mechanisms keep the learned hypergraph sparse and prevent combinatorial growth. Our sensitivity studies from Fig. 5 show that once $M$ is set to a reasonable value, hyperedge growth remains well-regulated in practice.
>
> ___
>
> **[Q2] Stability of learned hyperedges across folds or random initializations.**
>
> [A] In the revised manuscript, we added additional stability analysis to Appendix G to examine how consistently MASH learns hyperedges across folds and initializations. While individual incidence matrices cannot be aligned one-to-one across folds due to differences in training data, the overall structural patterns learned by the model remain highly consistent (Fig. 10). Across all initialization schemes, the incidence matrices display similar activation distributions, and the aggregated ROI-wise importance descriptions are nearly identical (Fig. 11). These findings show that MASH reliably converges to the similar hyperedge structure driven by data characteristics rather than random initialization. Consequently, the model consistently highlights the same meaningful ROIs and preserves robust group-level patterns across runs.
>
> ___
>
> **[Q3] Extension to temporal data.**
>
> [A] MASH is designed for a static brain graph with node-level features rather than raw fMRI time series. This is consistent with standard fMRI processing pipelines, where temporal signals are converted into functional connectivity matrices or summary statistics before analysis. Under this common setting, MASH can be directly applied to fMRI-derived functional networks, as demonstrated in our PPMI experiments. However, it does not explicitly model time-resolved changes in connectivity, which would require an extension beyond the current framework. We believe that this static functional representation remains widely used and clinically meaningful, and it provides a suitable way for evaluating MASH’s ability to model multi-scale high-order structure in fMRI-based brain graphs.

---

### Author Response · Authors · 2025-11-24
**General Answer**

We sincerely thank the reviewers for constructive reviews. We provide our answers to all the questions raised by the reviewers, which are updated in the manuscript (in **blue** text). We hope our answer addresses all the concerns and brings the paper up to acceptance.

---

> ### Author Response · Authors · 2025-11-28
>
> Dear Reviewers,
>
> Thank you again for your valuable feedback.
> We have provided our responses and would be happy to offer any additional clarifications if needed.
>
> Regards,
> Authors of submission 6625

---

### Author Response · Authors · 2025-11-29
**Summary Comment for AC**

Dear Area Chairs,

To assist with your meta-review, we briefly summarize how the paper was received in the reviews and how we have addressed the main concerns and feedback.

All four reviewers recognize that the proposed MASH framework is a **novel, theoretically grounded approach** that effectively combines graph wavelet–based multi-scale filtering with dynamic hypergraph learning with an application for brain network analysis. None of the reviews question the originality of the method; instead, the main questions and feedback focus on scalability, generalization to other datasets/disorders, further interpretability of learned hyperedges/ROIs, and the clarity, organization, and overall narrative flow of the paper and its experimental analysis.

In response to the reviewers’ questions and feedback, we have accordingly revised the main paper (highlighted in **blue**). The key updates include (but not limited to) the following:

- We added a detailed complexity and runtime/memory analysis and clarified that the number of hyperedges is explicitly bounded and kept sparse, demonstrating that MASH scales predictably with node count and the number of scales.

- We extended validation beyond ADNI and PPMI, including results on ABIDE (a multi-site autism cohort) and strict zero-shot cross-study generalization from PPMI to two independent PD cohorts, where MASH consistently outperforms strong brain-graph GNNs, transformer-based brain network models, and recent hypergraph baselines.

- We strengthened interpretability via statistical validation, showing significant associations between hypergraph-derived representations and various clinical scores, complemented by additional case studies of disease-relevant hyperedges and stability analyses across folds and random initializations that reveal robust ROI patterns.

- We clarified terminology and methodology (e.g., the notion of multi-scale high-order relationships, the roles of the main architectural components, and the choice of wavelet scales), updated the positioning relative to recent dynamic/learnable hypergraph methods (e.g., DHHNN, dwHGCN, and HyBRiD), expanded ablations and sensitivity studies (on initializations, the number of spectral filters, and learnable structures or weights for hyperedges), and restructured the Introduction, Related Work, and Methods sections based on the reviewers’ feedback to improve readability and logical flow.

As a whole, **these changes systematically address all weaknesses and questions raised in the reviews.** After these updates and our detailed rebuttal, there were no further public exchanges with the reviewers **although we tried to engage them**, and the scores remained at the pre-discussion values. As a result,  the current scores do not fully reflect the strengthened empirical evidence and clarified presentation. Given the reviewers’ clear agreement that the core idea is novel and theoretically solid, and in view of the additional experiments, statistical validation, and rewriting that directly address the points raised in the reviews, we believe the paper is now well structured and in good shap for publication. Please let us know if you have any questions.

---

### Meta-Review · Area_Chair_zSJE · 2026-01-09

**Summary:**

The reviewers acknowledged the strong innovation and theoretical depth of the proposed MASH framework, particularly its integration of multi-scale graph wavelet transforms with dynamic hypergraph learning to model high-order brain interactions, supported by convincing experiments and biological interpretability on ADNI and PPMI. With initial scores of 6, 4, 6, and 4 (borderline acceptance), the main concerns shaping my decision centered on clarity, robustness, and completeness rather than originality: reviewers highlighted ambiguities in method description and terminology, questioned computational complexity and scalability, and requested more comprehensive and up-to-date experimental comparisons, including recent dynamic hypergraph methods and stronger quantitative validation of ROI/hyperedge significance and biological relevance. If the authors can substantially improve clarity, strengthen scalability analysis, and expand and modernize experimental validation, the paper’s quality and impact could be raised to an acceptable level for acceptance.

**Reviewer Concerns:**

I believe the rebuttal effectively addresses the following issues:

- It addresses computational complexity and scalability: The authors added detailed complexity, runtime, and memory analyses, explicitly defining the sparsity and boundaries of the number of hyperedges, responding to concerns raised by XAdN and TYyY. This demonstrates the model's predictability as the number of nodes and scale increases, enhancing its practicality.

- It addresses experimental completeness and generalization: The authors expanded validation to include ABIDE (a multi-site autism cohort) and zero-shot cross-study generalization (from PPMI to two independent PD cohorts), showing that MASH outperforms GNN, Transformer, and recent hypergraph baselines. This directly addresses XAdN, TYyY, and dwqn's suggestions regarding more datasets and recent benchmarks, improving the method's robustness.

- It addresses interpretability and biological validation: The authors strengthened the interpretation of hyperedges/ROIs, including quantitative validation (such as comparisons with randomized hyperedges) and provided more biological insights (such as alignment with disease mechanisms). This responds to the requests of XAdN, TYyY, and dwqn, making the interpretation more convincing.

However, I believe the following issues remain or are not fully resolved:

- Clarity of Method Description: While the authors mention revising the paper, the rebuttal does not specify how to improve the vague description of the methods section (UYRR's concern) or clarify the terminology of "multi-scale" (TYyY). If the revisions are insufficient, this could still affect reader comprehension.

- Clear Differentiation from Existing Methods: dwqn and UYRR pointed out the insufficient localization of Transformers and recent dynamic hypergraphs. The rebuttal mentions updating related work but does not provide specific examples or analyses (such as the difference from dwHGCN). This may still require further discussion.

- Hyperedge Stability and Alternative Architectures: TYyY's questions regarding initial matrix generation and Transformer alternatives, as well as XAdN's tests on hyperedge stability, are not explicitly addressed in the rebuttal. This remains a potential weakness without the addition of relevant experiments.

Overall, the rebuttal demonstrates the authors' responsiveness, but certain clarity and detail issues need to be verified in the final version.

**Reviewer Scores:**

Reviewer XAdN (Initial 6): Their main concerns (computational cost, dataset limitations, hyperedge analysis) are directly addressed in the rebuttal, including complexity analysis, expanding the dataset, and strengthening interpretation. With participation in the discussion, they might affirm the adequacy of these improvements and raise the score to 7, as they have acknowledged the innovativeness, but the initial stance of "would not mind if rejected" provides flexibility.

Reviewer TYyY (Initial 4): Concerns included relationship definitions, missing comparisons, misleading terminology, and a lack of case studies. The rebuttal addresses these by expanding the benchmark and enhancing interpretability, but terminology clarification may still require discussion. With full participation, they might endorse the addition of biological cases and raise the score to 5, as the initial stance of "would not mind if accepted" provides neutrality, but confirmation that all issues are resolved is needed.

Reviewer UYRR (Initial 6): The focus is on the insufficient interpretation and methodological clarity of the Transformer. The rebuttal does not directly address these, but overall revisions (such as theoretical clarification) may indirectly help. If the authors provide detailed workflow and scale determination during the discussion, their score may remain at 6 because their confidence is lower (3), but they acknowledge theoretical depth.

Reviewer dwqn (Initial 4): Emphasizes abstract/introduction revisions, recent benchmarks, and quantitative interpretation. The rebuttal clearly updates relevant work, adds recent baselines, and provides quantitative validation. If involved, they may highly appreciate these changes and raise the score to 5 because they have praised the experiments and interpretation, but the initial weakness focuses on timeliness and positioning.

---

### Decision · Program_Chairs · 2026-01-26

Reject